# Use of Diagnostic Criteria from ACR and EU-TIRADS Systems to Improve the Performance of Cytology in Thyroid Nodule Triage

**DOI:** 10.3390/cancers13215439

**Published:** 2021-10-29

**Authors:** Davide Seminati, Giulia Capitoli, Davide Leni, Davide Fior, Francesco Vacirca, Camillo Di Bella, Stefania Galimberti, Vincenzo L’Imperio, Fabio Pagni

**Affiliations:** 1Department of Medicine and Surgery, University of Milano-Bicocca, Pathology, 20900 Monza, Italy; d.seminati@campus.unimib.it (D.S.); camillo.dibella@asst-monza.it (C.D.B.); vincenzo.limperio@gmail.com (V.L.); 2Bicocca Bioinformatics Biostatistics and Bioimaging B4 Center, School of Medicine and Surgery, University of Milano-Bicocca, 20900 Monza, Italy; giulia.capitoli@unimib.it (G.C.); stefania.galimberti@unimib.it (S.G.); 3Department of Radiology, ASST Monza, 20900 Monza, Italy; davide.leni@asst-monza.it (D.L.); davide.fior@asst-monza.it (D.F.); franscesco.vacirca@asst-monza.it (F.V.)

**Keywords:** ultrasound imaging, thyroid nodule, thyroid carcinoma, fine-needle aspiration

## Abstract

**Simple Summary:**

From a prospective series of 480 thyroid nodules, we compared the performances of the American College of Radiology (ACR) and the European Thyroid Association (EU) scoring systems in triaging thyroid nodules for fine-needle aspiration (FNA). FNA was recommended on 46.5% and 51.9% of the nodules using the ACR and EU-TIRADS scores, respectively. The ACR system demonstrated a higher specificity as compared to the EU-TIRADS (59.0% vs. 52.4%, *p* = 0.0012) in predicting ≥TIR3A/III (SIAPEC/Bethesda) nodules. Moreover, specific radiological features (i.e., echogenic foci and margins), combined with the cytological classes improved the specificity (97.5% vs. 91%, *p* < 0.0001) and positive predictive value (77.5% vs. 50.7%, *p* < 0.0001) of the cytology alone, maintaining an excellent sensitivity and negative predictive value.

**Abstract:**

**Objective:** The American College of Radiology (ACR) and the European Thyroid Association (EU) have proposed two scoring systems for thyroid nodule classification. Here, we compared the ability of the two systems in triaging thyroid nodules for fine-needle aspiration (FNA) and tested the putative role of an approach that combines ultrasound features and cytology for the detection of malignant nodules. **Design and Methods:** The scores obtained with the ACR and EU Thyroid Imaging Reporting and Data Systems (TIRADS) from a prospective series of 480 thyroid nodules acquired from 435 subjects were compared to assess their performances in FNA triaging on the final cytological diagnosis. The US features that showed the highest contribution in discriminating benign nodules from malignancies were combined with cytology to improve its diagnostic performance. **Results:** FNA was recommended on 46.5% and 51.9% of the nodules using the ACR and EU-TIRADS scores, respectively. The ACR system demonstrated a higher specificity as compared to the EU-TIRADS (59.0% vs. 52.4%, *p* = 0.0012) in predicting ≥ TIR3A/III (SIAPEC/Bethesda) nodules. Moreover, specific radiological features (i.e., echogenic foci and margins), combined with the cytological classes improved the specificity (97.5% vs. 91%, *p* < 0.0001) and positive predictive values (77.5% vs. 50.7%, *p* < 0.0001) compared to cytology alone, especially in the setting of indeterminate nodules (TIR3A/III and TIR3B/IV), maintaining an excellent sensitivity and negative predictive value. **Conclusions:** The ACR-TIRADS system showed a higher specificity compared to the EU-TIRADS in triaging thyroid nodules. The use of specific radiological features improved the diagnostic ability of cytology.

## 1. Introduction

In the international scenario, different alternative ultrasound (US) algorithms have been proposed for characterizing thyroid nodules [1]. The American College of Radiology (ACR) Thyroid Imaging Reporting and Data Systems (TIRADS) consists of a scale with increasing scores for specific US features of thyroid nodules and is able to stratify lesions with a progressively higher risk of malignancy (ROM) [2,3,4]. On this side of the Atlantic ocean, the European Thyroid Association (EU-TIRADS) proposed in 2017 a US pattern recognition method that combines high-risk criteria, such as nodule composition, echogenicity, margins, shape, and calcifications [5]. The main purpose of these scoring systems is the reduction in inappropriate fine-needle aspirations (FNAs), triaging thyroid nodules at their best. However, the final FNA results do not always match the pre-biopsy risk class assigned by these two systems. Indeed, the indication of a biopsy is influenced by subjectivity in US interpretation, different ROMs in the screened populations, and personalized clinical decisions [6,7]. On the other hand, defensive medicine, excessive scrupulousness, or limited experience could induce clinicians to perform FNA regardless of the TIRADS score, especially in large nodules, still depending on cytology for the final answer regarding the nature of a thyroid lesion and thus reducing the practical utility of TIRADS [8]. Although the most recent reports in the literature stress the recommendation to implement these algorithms in the clinical setting, the selection of more impactful radiological criteria may improve the contribution of these systems to risk assessments [6,9,10,11]. In this work, we present a comparative analysis of the most widely employed US classifications to support their systematic application in a first-level general hospital for the bioptic triage of thyroid nodules [8]. Moreover, the present paper supports the hypothesis that specific US features, coupled with cytological classes, might be better able predict the final malignant nature of a thyroid nodule.

## 2. Materials and Methods

### 2.1. Patients Selection

This prospective study included 448 consecutive patients who underwent 493 US-guided FNA from January to June 2019 at the interventional radiology clinic, ASST Monza, Italy, during an Italian Association for Research on Cancer (AIRC)-granted project for the diagnosis of thyroid carcinoma [8]. All nodules were subjected to FNA, regardless of their ACR/EU-TIRADS scores, after an endocrinological clinical indication. Thirteen lesions with an unsatisfactory cytology and no FNA repetition were excluded for a final set of 480 nodules and 435 subjects. Histology was available in 49 resected nodules. US was performed at the 12-month follow-up visit in 352 patients, while 79 cases, corresponding to 70 TIR3A/III, 8 TIR3B/IV, and 1 TIR4/V, were lost to follow-up. This study was approved by the ASST Monza Ethical Board (October 2016, 27102016) and appropriate informed consent was obtained from all patients.

### 2.2. Ultrasound Evaluation

Patients were placed in supine position with their neck in hyperextension. The US was performed with the Philips Epiq Elite machine. For each nodule, the radiologists used real-time clips to measure the major axis and analyzed its composition, echogenicity, shape, margins, and the presence of calcifications, as previously described [3]. The final theoretical indication for FNA execution was formulated according to the ACR and EU-TIRADS algorithms [3,5]. US was performed by three different radiologists (DL, DF, FV) who were experts in thyroid imaging and who contributed equally to the evaluation of the case series.

### 2.3. Cytopathology and Histopathology

Aspiration was performed by two pathologists (FP and CDB) with 10 years of experience in thyroid FNA under US guidance with 22–25 gauge diameter needles. The aspirated material was smeared onto 3–4 traditional slides per nodule. The slides were fixed with spray alcohol (Cytofix, propan-2-ol) and then stained with Papanicolau, or air-dried and stained with May–Grunwald Giemsa. Cases were diagnosed according to two standard systems for reporting thyroid cytopathology: the Italian Society of Pathology classification (SIAPEC) and the Bethesda System [12,13]. The TIRADS indication of FNA was considered correct in the presence of a cytology ≥TIR3A/III (SIAPEC/Bethesda). TIR1c-TIR2 patients and those with TIR3 who did not undergo surgery, underwent a US examination 12 months after the first US-guided FNA performed by the same radiologist [14]. Nodules were considered benign in the absence of the following conditions:new echographic suspicious features—i.e., US features with ≥2 ACR-TIRADS points;>20% increase in size;enlarged lymph nodes;appearance of new suspicious nodules—i.e., with ACR or EU-TIRADS class ≥ 3.

A histological evaluation was performed on the surgical specimens of total or hemi-thyroidectomy [15]. The tissue was formalin-fixed, paraffin-embedded, and stained with haematoxylin and eosin. The ROM was estimated according to histological evaluation or US follow-up examination.

### 2.4. Statistical Analysis

Mean and standard deviation or quartiles were used for descriptive purposes, as appropriate. The diagnostic ability of the TIRADS systems in distinguishing thyroid nodules that required or not FNA was evaluated using cytology as reference standard (<TIR3A/III vs. ≥TIR3A/III) in all 480 nodules. The results of the TIRADS classifications and cytology were also compared to the histopathological/follow-up findings on 401 nodules. Sensitivity, specificity, and positive and negative predictive values (PPV and NPV) were calculated alongside their 95% confidence intervals (CI). The McNemar test was considered for the comparison of the performances (two-sided test, α = 0.05). A decision tree was applied to the cytological classes and to the single TIRADS ecographic components of each thyroid nodule to explore whether and which US features were relevant in malignancy detection. The risk of malignancy was calculated as the rate of prevalence. All of the statistical analyses were performed using the open-source R software v.3.6.0 (R Foundation for Statistical Computing, Vienna, Austria).

## 3. Results

### 3.1. ACR vs. EU-TIRADS

The comparison between the ACR and EU systems in classifying the 480 nodules is shown in Table 1.

Considering the nodules requiring FNA as per the ACR (*n* = 223, 46.5%) and EU-TIRADS (*n* = 249, 51.9%) criteria, 86 (38.6%) and 90 (36.1%) were deemed to be ≥TIR3A/III after cytological assessment. Overall, a good agreement between the two systems was noted, reaching an accordance for FNA indication in 87.1% (418/480) of the nodules. The major discrepancies on the execution of cytology were observed in the setting of EU-TIRADS class 3, with 26 cases that would not be submitted to biopsy as per ACR, either for different US scores (*n* = 22) or size cut-off (*n* = 4). Similarly, 12 cases that received FNA following the ACR-TIRADS criteria did not reach the indication in the EU-TIRADS system, either for different US scores (*n* = 1) or size cut-offs (*n* = 11). Finally, 11.1% (6/54) of the cases labeled as class 5 in both the systems would be biopsied only following ACR due to the presence of a size “grey-zone” of exactly 1 cm in the classifications [3,5]. The assessment of the performances of these systems in the final FNA indication showed a significantly higher specificity for the ACR as compared to the EU-TIRADS (59.0% vs. 52.4%, *p* = 0.0012), with a similar sensitivity (58.9% vs. 61.6%, *p* = 0.3173), PPV (38.6% vs. 36.1%, *p* = 0.1116), and NPV (76.7% vs. 75.8%, *p* = 0.5288) (Table 2 and Table 3). The difference noted in terms of the specificity was mainly due to the discordances of the two systems in terms of the size cutoff and the US feature “echogenicity”.

Histological evaluation was performed on 10.2% (49/480) of the nodules, 81.6% (40/49) of which had an FNA result ≥ TIR3B/IV. A total of 34 out of the 49 (69.3%) nodules showed malignancy (27 PTC, 3 EFVPTC, 1 Hurthle cell carcinoma, 1 medullary carcinoma, 1 anaplastic carcinoma, and 1 metastasis of the melanoma). The ACR-TIRADS confirmed a significantly higher specificity (57.2% vs. 51.2%, *p* = 0.0019), with a similar sensitivity (67.6% vs. 70.6%, *p* = 0.6547), PPV (12.8% vs. 11.8%, *p* = 0.4030), and NPV (95% vs. 94.9%, *p* = 0.9435) in the diagnosis of malignancy (Table 4). As reported for the analysis of FNA indication, the difference in terms of the specificity between the two systems was mainly driven by the size cutoff and US feature “echogenicity”.

### 3.2. Combining US Features with Cytology for Diagnostication

The diagnostic performance of cytology alone, using the TIR3A/III class as the threshold, showed a sensitivity of 100% (34/34, 95% CI = 89.7–100%), a specificity of 91% (334/367, 95% CI = 87.6–93.7%), a PPV of 50.7% (34/67, 95% CI = 38.2–63.2%), and an NPV of 100% (334/334, 95% CI = 98.9–100%) (Table 4).

As expected, combining the TIRADS and cytology classes did not improve their performance. However, a decision tree applied to cytology and single US features of the two TIRADS systems identified two independent characteristics, i.e., echogenic foci and irregular margins, able to contribute to diagnostication, especially in the setting of undetermined nodules (Figure 1).

Indeed, this approach adequately detected four cases of carcinoma, otherwise classified as TIR3A/III (*n* = 1) and TIR3B/IV (*n* = 3). Moreover, three indeterminate nodules (one TIR3A/III and two TIR3B/IV) were wrongly classified as benign by the application of this approach and corresponded to an encapsulated follicular variant of papillary thyroid carcinoma (EFVPTC), whose malignant potential is still debated. Similarly, among the eight cases misclassified as malignant by the decision tree, four were diagnosed as follicular adenomas on the final histology (two TIR3A/III and two TIR3B/IV), and, although these lesions are benign, they usually require the complete histological assessment of the final surgical specimen for the exclusion of carcinoma. Finally, the adoption of this combined US/cytology approach led to a significant increase in the specificity (97.5% vs. 91%, *p* < 0.0001) and PPV (77.5% vs. 50.8%, *p* < 0.0001), with still excellent values of sensitivity (91.2% vs. 100%, *p* = 0.0833) and NPV (99.2% vs. 100%, *p* = 0.0820), as compared to cytology alone (Table 4).

## 4. Discussion

Various studies have shown the good performance of ACR-TIRADS in selecting thyroid nodules deserving FNA, stressing its high “rule-out” role, as confirmed by our group here and previous experience [1,6,8,16,17,18]. In the present work, the ACR and EU-TIRADS systems recommended FNA in 46.5% (223/480) and 51.9% (249/480) of the nodules, respectively. The rate of correct FNA indication as per ACR and EU (i.e., nodules with indication for FNA and with a cytology result ≥TIR3A), was 38.6% (86/223) and 36.1% (90/249), respectively, without statistical significance (*p* = 0.5872), in accordance with the literature [1,19]. On the other hand, in this experiment, the proportions of cytological tests that could be avoided as per ACR-TIRADS and EU-TIRADS were 28.5% (137/480) and 33.1% (159/480), close to what has been recently reported in a meta-analysis (25% and 38%, respectively) [19]. Moreover, some studies have also found no statistically significant differences in the pooled diagnostic performances between the two scores [20].

A meta-analysis showed a pooled sensitivity and specificity of 95% and 55% for TIR4/V and TIR5/VI classes in the ACR system, respectively, and a pooled sensitivity and specificity of 96% and 52% for the same classes in the EU-TIRADS [21]. Another recent meta-analysis showed better overall diagnostic performances for ACR than for EU (sensitivity 74% vs. 54%, specificity 64% vs. 53%, PPV 43% vs. 29%, NPV 84% vs. 81%, respectively) [22]. Other reports indicate that the ACR-TIRADS had significantly higher specificity and PPV but a lower sensitivity and similar NPV when compared to the EU-TIRADS system [1,10,23]. Our findings are in line with those of the latter studies, showing a slightly lower sensitivity (58.9% vs. 61.6%, *p* = 0.3173) but a significantly higher specificity (59.0% vs. 52.4%, *p* = 0.0012) for ACR, with similar PPV (38.6% vs. 36.1%, *p* = 0.1116) and NPV (76.7% vs. 75.8%, *p* = 0.5288). Although the cytological classes already help in the stratification of patients through a predicted ROM known for every single class, other ancillary tests/criteria might help in the distinction of lesions with malignant behavior, especially in indeterminate categories [12,13,24,25]. According to the routine cytopathological classifications, the expected rates of malignancy for classes TIR3A/III and TIR3B/IV are <10%/5–15% and 15–30%, respectively [12,13]. The results of an Italian study showed a malignancy rate estimated based on surgical outcomes of 17% and 40% for TIR3A/III and TIR3B/IV, respectively [26]. In our series, based on the cases with available surgical excision, the risk of malignancy was 22% (2/9) and 38% (5/13) in the TIR3A/III and TIR3B/IV classes, respectively, which is slightly lower compared to those reported in a recent meta-analyses (TIR3A/III 10%, TIR3B/IV 52%) [27]. Wu et al. tried to better stratify the indeterminate nodules through a KRAS mutation assessment by polymerase chain reaction (PCR), assuming that this genetic alteration is usually associated with a moderate risk of malignancy, mainly represented by follicular tumors with a good prognosis [24]. However, this approach failed to improve the diagnostic performances of the sole ACR-TIRADS, and the KRAS mutation was exclusively found in tumors classified as TIR3B/IV, with no malignant mutated cases in the TIR3A/III class. In this setting, the combination of the existing radiological scales (e.g., ACR and EU-TIRADS) with the routinely used cytological systems has been previously investigated in challenging nodules. Hong et al. combined ultrasound patterns with cytology, finding a lower risk of malignancy for TIR3A/III class nodules with a Korean TIRADS 3 score [28]. A meta-analysis evaluated the putative role of thyroid US in predicting the malignancy of TIR3A/III nodules, finding a high variability in terms of the sensitivity and specificity among the studies analyzed, probably due to the heterogeneity of the different US criteria employed to detect malignant nodules and due to the variable prevalence of malignancies in the different cohorts [7]. However, the only feature with a significant influence on diagnostic accuracy was the increased vascularization of the nodules, which is not taken into account by both the ACR and EU-TIRADS systems. Other recent studies investigated the impact of different dimensional cutoffs, e.g., ≤2 and >2 cm, on the final performances of the available US systems, showing a range of values quite close to the ones obtained in the present cohort with the ACR and EU-TIRADS systems [8,29]. An innovative approach was proposed in 2017 by He et al., creating a new algorithm that significantly increased the predictive performance of US features [30]. In our case series, we found no improvement in diagnostic accuracy by combining either ACR or EU-TIRADS and the cytology class. Nevertheless, the combination of the cytological classes with specific US features extracted from the ACR system—namely, echogenic foci and margins with a not-zero score—led to a significant increase in specificity and PPV, with a slight reduction in sensitivity and NPV as compared to cytology alone (Table 4). The introduction of this new combined US-cytological approach allowed the correct identification of nodules with a ROM > 60%, which could certainly benefit from surgery, as well as those with a low ROM (<10%), still amenable for clinical follow-up as per SIAPEC and Bethesda operative indications (Figure 1) [12,13]. Although these promising results might represent a starting point for the improvement of the actual diagnostic performances of cytological classifications, we recognize some limitations in the present study: the limited number of cases with indeterminate cytology, i.e., TIR3A/III and TIR3B/IV, the short US follow-up period (12 months) for nodules who did not undergo surgery, the low number of cases that underwent surgery, and the high prevalence of benign nodules. This latter situation reflects the target population of multinodular hyperplastic goiters typical of a first-level general hospital population, which only partly reflects the settings encountered in highly specialized centers with a large proportion of malignant cases and where molecular testing may be more easily performed [31]. However, as a description of a real-life practice in first-level general hospitals, this could be of help in validating the proposed combined approach in larger cohorts to further verify the reported diagnostic performances, eventually leading to their implementation for the clinical assessment of thyroid nodules.

## 5. Conclusions

In this study, we compared the performances of the ACR and EU-TIRADS systems, showing the statistically significant higher specificity of the ACR and a comparable sensitivity, PPV, and NPV. The coupled use of the ACR or EU-TIRADS with the cytological class did not improve the diagnostic performance in identifying malignant nodules. However, the combination of specific ultrasound features with cytological class might help in the diagnosis of malignant nodules, especially with an indeterminate for the malignancy pathological report.

## Figures and Tables

**Figure 1 cancers-13-05439-f001:**
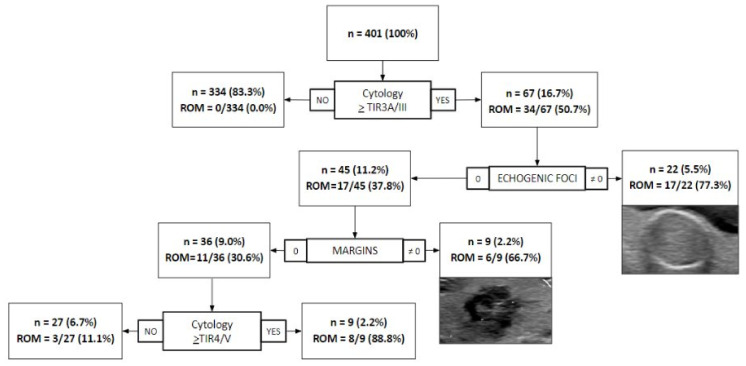
Nodules’ risk of malignancy (ROM, based on histology or follow-up) using a combination of cytology with ultrasound ACR features of the echogenic foci and margins.

**Table 1 cancers-13-05439-t001:** Comparison between the ACR and EU-TIRADS systems in the indication to perform FNA.

	ACR-TIRADS								
	No Indication of FNA					Indication of FNA			
EU-TIRADS	ACR1	ACR2	ACR3 <2.5 cm	ACR4 <1.5 cm	ACR5 <1.0 cm	ACR3 ≥2.5 cm	ACR4 ≥1.5 cm	ACR5 ≥1.0 cm	Total
No indication of FNA									
EU 1	0	0	0	0	0	0	0	0	0
EU 2	30	2	0	0	0	0	0	0	32
EU 3 ≤ 2.0 cm	0	15	50	3	0	0	1	0	69
EU 4 ≤ 1.5 cm	0	0	19	81	0	0	11	0	111
EU 5 ≤ 1.0 cm	0	0	0	10	3	0	0	6	19
Indication of FNA									
EU 3 > 2.0 cm	0	22	4	0	0	39	9	0	74
EU 4 > 1.5 cm	0	0	8	0	0	16	69	0	93
EU 5 > 1.0 cm	0	0	0	10	0	0	27	45	82
Total	30	39	81	104	3	55	117	51	480

**Table 2 cancers-13-05439-t002:** ACR-TIRADS vs. SIAPEC/Bethesda system cytological classifications. False negative cases are in bold. False positive cases are in italics.

	SIAPEC/Bethesda System						
ACR-TIRADS	TIR1c/I	TIR2/II	TIR3A/III	TIR3B/IV	TIR4/V	TIR5/VI	Total
**No indication of FNA**							
**ACR 1**	12	16	**2**	**0**	**0**	**0**	30
**ACR 2**	1	33	**5**	**0**	**0**	**0**	39
**ACR 3 < 2.5 cm**	0	63	**13**	**2**	**2**	**1**	81
**ACR 4 < 1.5 cm**	0	72	**21**	**4**	**3**	**4**	104
**ACR 5 < 1.0 cm**	0	0	**2**	**0**	**1**	**0**	3
**Indication of FNA**							
**ACR 3 ≥ 2.5 cm**	*1*	*35*	15	3	1	0	55
**ACR 4 ≥ 1.5 cm**	*1*	*77*	29	7	3	0	117
**ACR 5 ≥ 1.0 cm**	*0*	*23*	9	5	2	12	51
**Total**	15	319	96	21	12	17	480

**Table 3 cancers-13-05439-t003:** EU-TIRADS vs. SIAPEC/Bethesda system cytological classifications. False negative cases are in bold. False positive cases are in italics.

	SIAPEC/Bethesda System						
EU-TIRADS	TIR1c/I	TIR2/II	TIR3A/III	TIR3B/IV	TIR4/V	TIR5/VI	Total
**No indication of FNA**							
**EU 1**	0	0	**0**	**0**	**0**	**0**	0
**EU 2**	12	18	**2**	**0**	**0**	**0**	32
**EU 3 ≤ 2.0 cm**	0	54	**11**	**1**	**2**	**1**	69
**EU 4 ≤ 1.5 cm**	0	84	**21**	**3**	**2**	**1**	111
**EU 5 ≤ 1.0 cm**	0	7	**7**	**2**	**2**	**1**	19
**Indication of FNA**							
**EU 3 > 2.0 cm**	*2*	*47*	20	4	1	0	74
**EU 4 > 1.5 cm**	*1*	*67*	20	3	2	0	93
**EU 5 > 1.0 cm**	*0*	*42*	15	8	3	14	82
**Total**	15	319	96	21	12	17	480

**Table 4 cancers-13-05439-t004:** Diagnostic performances of TIRADS systems, cytology, and combination of US features and cytology in the diagnosis of malignant nodules (*n* = 401).

	TP	FP	TN	FN	Sensitivity (%)	Specificity (%)	PPV (%)	NPV (%)
**ACR**	23	157	210	11	67.6 (23/34)	57.2 (210/367)	12.8 (23/180)	95.0 (210/221)
					[49.5–82.6]	[52.0–62.3]	[8.3–18.6]	[91.3–97.5]
**EU**	24	179	188	10	70.6 (24/34)	51.2 (188/367)	11.8 (24/203)	94.9 (188/198)
					[52.5–84.9]	[46.0–56.4]	[7.7–17.1]	[90.9–97.6]
**Cytology**	34	33	334	0	**100** (34/34)	91.0 (334/367)	50.7 (34/67)	**100** (334/334)
					[89.7–100]	[87.6–93.7]	[38.2–63.2]	[98.9–100]
**Cytology +EF/M**	31	9	358	3	91.2 (31/34)	**97.5** (358/367)	**77.5** (31/40)	99.2 (358/361)
					[76.3–98.1]	[95.4–98.9]	[61.5–89.2]	[97.6–99.8]

The best performances are in bold, 95% confidence intervals in brackets. TP: true positive; FP: false negative; TN: true negative; FN: false negative; PPV: positive predictive value; NPV: negative predictive value; EF/M: echogenic foci/margins.

## Data Availability

The data presented in this study are more extensively available in Leni, D.; Seminati, D.; Fior, D.; Vacirca, F.; Capitoli, G.; Cazzaniga, L.; Di Bella, C.; L’Imperio, V.; Galimberti, S.; Pagni, F. Diagnostic Performances of the ACR-TIRADS System in Thyroid Nodules Triage: A Prospective Single Center Study. Cancers 2021, 13, 2230. https://doi.org/10.3390/cancers13092230.

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
