# Peer review of "Use of Diagnostic Criteria from ACR and EU-TIRADS Systems to Improve the Performance of Cytology in Thyroid Nodule Triage"

_cancers, 2021, doi:10.3390/cancers13215439_

Round 1
Reviewer 1 Report
In prospective study entitled: “Diagnostic criteria from ACR and EU-TIRADS systems to improve the performances of cytology in thyroid nodules triage”, the authors compared ACR and EU-TIRADS scoring systems in thyroid nodules. They found interesting results for Bethesda III lesions.Although several meta-analyzes comparing the two classifications have been published, the combined assessment of the morphological features of the lesions with the result of the cytological examination is an original approach.
The work, however, requires clarification in several aspects:
- Line 71-please precise the US appearance, TIRADS(which?) was taken into account.
Line 85-how is the interobserver agremment, whether the authors performed such a compatibility analysis between researchers on a small subset
- According the results presented in 75-78 line, at the end of the study should be added, about the limitation resulting from cytology results comparing to histology.
- Line 99/100-precise the features…
There are few metanalysis, in which authors compared the TIRADS classification including ACR and EUTIRADS. It would be worthwhile to refer to the meta-analyzes listed below.
- Yang R, Zou X, Zeng H et al. Comparison of Diagnostic Performance of Five Different Ultrasound TI-RADS Classification Guidelines for Thyroid Nodules. Front Oncol 2020; 16: 10: 598225. doi: 10.3389/fonc.2020.598225.
- Liu H, Ma AL, Zhou YS et al. Variability in the interpretation of grey-scale ultrasound features in assessing thyroid nodules: A systematic review and meta-analysis. European Journal of Radiology 2020; 129: 109050. doi: 10.1016/j.ejrad.2020.109050.
- Castellana M, Castellana C, Treglia G et al. Performance of five ultrasound risk stratification systems in selecting thyroid nodulesfor FNA. J Clin Endocrinol Metab 2020; 105: dgz170. doi: 10.1210/clinem/dgz170.
.
- Kim PH, Suh ChH, Baek JH et al. Eu Radiol 2021; 31: 2877-2885. Unnecessary thyroid nodule biopsy rates under four ultrasound risk stratification systems: a systematic review and meta-analysis doi: 10.1007/s00330-020-07384-6.
Reviewer 2 Report
It is very interesting study. However, this type of studies have limitations in that accurate results cannot be known without final surgery in patients with moderate or higher risk.
I would like to address a few comments or suggestions to increase the completeness of this study.
Question 1)
If possible, it would be nice to compare the two systems by including the AACE/ACE/AME system and the modified K-TIRADS system in addition to the pre-existing two systems.
Question 2)
Although this study is a prospective study, in retrospect, was there any unnecessary FNA?
According to the recent meta-analysis (Eur Radiol. 2021 31(5):2877-2885), the pooled unnecessary biopsy rates of ACR-TIRADS, ATA, EU-TIRADS, and K-TIRADS were 25%, 51%, 38%, and 55%, respectively. Therefore, it concluded that the pooled unnecessary biopsy rate of ACR-TIRADS was significantly lower than that of ATA (p < .001) and K-TIRADS (p < .001), and also lower than that of EU-TIRADS, but not reaching statistical significance (p = .087).
Question 3)
What exactly does ‘echogenic foci’ mean? Perhaps it means calcification within the tumor.
Is that right?
Question 4)
Is there a difference in sensitivity and specificity, diagnostic odds ratio depending on the size of the tumor? According to the recent study (for example, Ultrasonography 2021 40(4):474-485), in small thyroid nodules (≤2 cm), ACR TIRADS had the highest reduction rate of unnecessary biopsies (76.3%) and the lowest sensitivity (76.1%). The modified K-TIRADS had the second highest reduction rate of unnecessary biopsies (67.6%) and sensitivity (86.6%). In addition, the modified K-TIRADS and ACR TIRADS had the highest diagnostic odds ratios (P=0.165). However, in large nodules (>2 cm), the sensitivity of the ACR TIRADS for malignancy was significantly lower (88.8%) than the sensitivities of the modified K-TIRADS and other systems, which were very high (98.7%-99.3%) (P<0.001).
